# Flood exposure of environmental assets

Authors:
Gabriele Bertoli[1], Chiara Arrighi[1], Enrica Caporali[1]

1: Department of Civil and Environmental Engineering, University of Florence, via di Santa Marta, 3. Firenze (Italy)

Corresponding author: Chiara Arrighi. E-mail: chiara.arrighi@unifi.it

## Abstract

Environmental assets provide important benefits to society and support the equilibrium of natural processes. They can be affected by floods, nevertheless, flood risk analyses usually neglect environmental areas due to (i) a lack of agreement on what should be considered as an environmental asset, (ii) a poor understanding of environmental values, and (iii) the absence of damage models. The aim of this work is to advance the understanding of environmental exposure to river floods by first identifying asset typologies that could be considered in flood risk analyses and second, by introducing a method, named EnvXflood, to estimate flood exposure qualitative values of environmental assets. The method is structured around three levels of detail requiring increasing information, from a fast and parsimonious analysis suitable for regional assessment to a detailed ecosystem-service-based site analysis. Exposure focuses on the social and environmental value of the assets. Social values were investigated by means of a participatory approach. The method was tested on three case studies in Italy (Tuscany region, Chiana, and Orcia basins). The Ecosystem Services weighting obtained from the participatory approach highlights the perceived leading importance of the biodiversity-supporting service. The results of the analyses show that the environmental assets related to water, such as rivers, lakes, and wetlands, are the assets most exposed to floods. Notwithstanding, commonly they are not considered as exposed assets in the usual river management practices. Further research should aim at consolidating the asset typologies to be included in environmental exposure analysis and their social and ecological value, moving towards a coherent understanding of environmental flood impacts.

1. Introduction

Environmental assets are crucial for human life, the vitality of ecosystems, and the equilibrium of natural processes. Environmental assets, broadly, are all the naturally occurring entities "including those which have no economic values, but bring indirect uses benefits, options and bequest benefits or simply existence benefits which cannot be translated into a present-day monetary value" (United Nations, 1993). Among the natural hazards that can impact the environment, river floods have been reported, in the aftermath of recent events, to have affected water resources and water related ecosystems (Arrighi and Domeneghetti, 2024). Floods influence on environmental assets and their ecosystems, in general, can be expressed as the temporary or permanent alteration of the capability of providing ecosystem services. In detail, one of the main concerns is the pollutants transportation by floodwaters (Arrighi et al., 2018), (Thieken et al., 2016) which also might increase contaminants concentration in fishes (Ondarza et al., 2012; Stewart et al., 2003), and destroy habitats (Aldardasawi and Eren, 2021). A recent field study, demonstrated that flooding cause more severe and lasting effects on ecosystem processes, including plant productivity and nutrient cycling, compared to droughts (Dodd et al., 2023). Moreover, flooding can cause damages due to the enhanced sediment transport (Weber et al., 2023), (Kelman and Spence, 2004) and impacts on aquatic and terrestrial life from temporal turbidity and water quality alteration (Caballero et al., 2024). Floods are reported to impact also on food production (Pacetti et al., 2017), breaching in and altering the riparian zones (Guan et al., 2015), and modifying plants reproduction and tree survival (Fischer et al., 2021; Predick et al., 2009), among others. Nevertheless, potential flood impacts on environmental assets are difficult to understand and (Thieken et al., 2016). In fact, for some ecosystems, flooding may represent a regulating natural phenomenon (Natho, 2021), which provides certain habitats with organic and inorganic matter and ensures sustainability and the preservation of biodiversity (Physiological-Ecological Impacts of Flooding on Riparian Forest Ecosystems, 2022). There, the concern is e.g. when due to anthropogenic pressures, floodwaters transport or resuspend undesired substances, e.g., contaminants originating from human activities (Barber et al., 1998; Petty et al., 1998; Weber et al., 2023). Assessing the flood exposure of the environmental assets turns out to be useful in many different applications and studies, whether they are aimed at assessing the vulnerability of the assets or aimed at assessing potential positive effects of the floods on such natural assets, also taking in account that human activities can strongly influence the flood regulating capacity of environmental assets (Mori et al., 2021).

The present work is intended to be potentially applied in different areas, but it is developed with the aim to provide an effective instrument for researchers and professionals to fulfil the European requirements in matter of flood risk assessment.
Indeed, the European Flood Directive requires assessing the potential adverse consequences of floods on the environment and preventing and reducing these impacts. The term *environment* broadly includes all uses of land from urban to agricultural ones, and the natural environment. Henceforth the term environment will refer to the natural environment.

Risk is the probability of a loss, and one of the most widely accepted definition is based on three elements (David Crichton, 1999) i.e., the hazard H, which is a process or a phenomenon threatening the elements at risk, the exposure E to the hazard, describing the value and location of the elements at risk, and the vulnerability V, that is the extent to which the elements at risk will suffer of damage or loss (David Crichton, 1999).
For assessing flood risk of environmental assets, given that flood hazard analyses are managed by the water authorities and sufficiently detailed for this purpose, one of the most important steps forward is to better describe their exposure to floods. The next step is the vulnerability assessment, which, however, is not covered in this study.

The exposure is commonly quantified by the *value or number* of assets located in the flooded area (Kron, 2005). Some frequently adopted exposure metrics are the resident population, the number of affected economic activities, the footprint area of the buildings, and their monetary value (Kang et al. 2005), or their replacement value (Amadio et al., 2016; Wu et al., 2019; Ye et al., 2019). No standard describing metrics are now commonly accepted and available for the environmental heritage and assets, except for their area, and most of the exposure assessments only report if the asset lies in a floodable site or not. Moreover, there is no standard agreement on which environmental assets must be included in flood risk management plans. It is believed that the evaluation of environmental assets needs a new approach from the researchers (Guijarro and Tsinaslanidis, 2020) aimed at including new elements in the valuation process.
Currently, the environmental valuation is usually obtained following different economic instruments, although not exhaustive (Gómez-Baggethun and Muradian, 2015; Venkatachalam, 2004)
It can be exploited through the Total Economic Value (TEV) approach, but the specific characteristics of each environmental asset do not allow a uniform treatment with the TEV model (Guijarro and Tsinaslanidis, 2020).
Other economic metrics usually applied to the environmental evaluation and similar assets (such as the cultural

heritage) are the "contingent evaluation" method, which encompasses both the "willingness to pay" and the "willingness to accept" approaches (Venkatachalam, 2004), as well as the "travel cost" method. These methods can eventually be integrated in the final evaluation of environmental assets, but only as indicators, because they are not able to fully represent the complexity of the environmental assets. Issues are also related to the spatial scale of the evaluation, because those methods are mainly applicable to small-scale and site-specific studies, but flood risk analyses often are conducted at the watershed or regional scales.

Environmental assets are jointly tangibles and intangibles assets, due to their physical and technical values combined with their cultural, aesthetic, and spiritual values, adding more challenging questions in their proper evaluation. Some experiments to apply a "commodification" of these aspects have been explored (Angeli Aguiton, 2020) but it is believed that the monetization of all the different typologies of environmental assets is utopistic and not representative of the reality.

The intangible value also introduces a spatial and temporal variability of the estimate because it is strictly related to the social context and time in which the asset is evaluated.

The study performed by Robert Costanza (Costanza et al., 1997) and published as "The value of the world's ecosystem services and natural capital", which is one of the cornerstones in understanding the value of the environment, makes clear that it is crucial to also focus on the analysis of the *ecosystem services* that the natural environment can provide to human life. Ecosystems are defined as "a *dynamic* complex of plant, animal and micro-organism communities and their non-living environment interacting as a functional unit" by the Convention on Biological Diversity (UN, 1996). Ecosystem services can be defined as "the conditions and processes through which natural ecosystems, and the species that comprise them, sustain and fulfil human life" (Ecosystems and their services, 2022). As stressed by Costanza (Costanza et al., 1997), "ecosystem services are largely outside the market", and this elucidates that an approach not closely centred in economic value could be developed and weighted, aiming at providing an evaluating framework that goes beyond the market, and which is based on the social and natural value of the environment, which, indirectly, also include the economic aspect. Moreover, despite the diversity of nature's values, most policymaking approaches have prioritized a narrow set of values at the expense of both nature and society, as well as of future generations, generally considering only those values of nature reflected through markets and not accounting for the over-exploitation of nature, its ecosystems and biodiversity, and the impact on long term sustainability (IPBES, 2022).

Examples of studies that identify and assess flood exposure of natural assets (Andrew Tait, 2019) are rarely found in the literature especially when dealing with larger territorial scales, as regional or river basin scales, more typical of risk management plans.

The present work aims at advancing the current state of the art in the assessment of flood exposure of environmental assets, with the following specific objectives: (i) develop a taxonomy for environmental assets exposed to flooding, (ii) develop a new non-monetary method for valuing the environmental assets able to differentiate among asset typologies, (iii) propose a spatial index of environmental exposure that can support river district Authorities in flood risk mapping and management.

The method here proposed will be tested and applied to a case study in Italy, where the Italian law (Legislative Decree 49/2010) specifically asks to evaluate and manage the flood risk for the environmental assets and to produce flood risk maps for a list of assets, including the environmental assets in the areas potentially exposed to floods, but large subjectivity is left in the identification of the assets.

This is a starting point in enhancing the representation of the environmental assets while analysing flood risk, also contributing to a more informed risk evaluation, and consequently to a better risk management.

2. Materials and methods

2.1. Environmental assets identification and taxonomy

To fulfil the objective (i), first step consists of the research and selection of the assets to be included in the analysis of environmental exposure. In fact, given the diversity of environmental assets and their level of protection, a unique spatial database does not exist and must be created *ad-hoc* by collecting information from different sources. The work starts from the definition provided by UNESCO of natural heritage as "natural places in the world, characterized by their outstanding biodiversity, ecosystems, geology or superb natural phenomena". But the aim of the work is to consider the meaning of "environmental asset" in its broader connotation, as suggested by the definition reported in the OECD Glossary of Statistical terms (OECD, 2008), together with the one provided by the United Nations, which consider all the naturally occurring entities "including those which have no economic values, but bring indirect uses benefits, options and bequest benefits or simply existence benefits which cannot be translated into a present day monetary value" (United Nations, 1993). Thus, here are considered as environmental

assets also the sites which characterize the natural and cultural heritage (mixed sites), the landscape, the natural resources, the activities, the history, and the climate of a country, or of a specific location. Those assets define and influence the characteristics, opportunities, shape, and well-being of the neighbouring human settlements and activities. Most of the environmental assets are identified by international, national or regional laws, which we used as identification and classification instruments. This approach facilitates the standardization of the procedure over different areas, and allows to catch all the most relevant assets, potentially not including some minor, local assets, which may not be protected or identified by the laws. This is in line with the objectives of the present study, especially regarding international, national, and regional scale applications, since minor and less relevant assets have, by definition, less value, with expected low impacts on the final exposure assessment. In case of studies conducted at catchment scale, or even more local scales (e.g. municipality), specific investigation on the local peculiarities and assets is still suggested, also depending on the capillarity of the local legislation. After identifying the assets commonly protected from international to local levels, a classification of environmental assets has been set, providing a systematic framework for categorising and understanding the different natural features that may be exposed to floodings. The assets have been grouped according to macro characteristics and ecosystem typology, enabling a more organized approach to their identification. The different geometric entities required to describe environmental assets in a geographical information system pose an additional challenge in quantifying their exposure with synthetic indices. All the assets identified for the case studies were collected and represented in a GIS environment with different geometric features, as:

-polygons, in case of a large portion of territory, such as a forest or a wetland;
-lines, in the case of networks, such as rivers or naturalistic itineraries;
-points, for localized assets, such as a monumental tree or a water spring.

## 2.2. EnvXflood Model structure and levels of analysis

The environmental exposure analysis of the EnvXflood method here introduced is designed to assess the exposure to floods of environmental assets, capturing and qualitatively expressing their value, following objective (ii) of our study. The model has a flexible architecture, to be adaptable to different contexts, and to be easily integrated with the typical workflows involved in geospatial analysis, with the use of Geographic Information System (GIS) and spreadsheets. The core of the estimation framework is the identification, and the subsequent evaluation of objective characteristics recognized to belong to the asset, avoiding direct focus on the economic aspect, instead favouring the ecosystem and social value. The method works both with the legislative framework and with Ecosystem Services delivered by the identified assets. Ecosystem services are powerful instruments capable to describe the *natural capital* and its relations with the human being and its activities (Chen et al., 2022; Liu et al., 2024), recently gaining a growing interest and consideration from the scientific community. After the identification and classification of the asset, the following step regards the weighting of the features attributed to each asset. Among the results, there is the overall Environmental Exposure Index (*EEI*), as detailed in the following paragraphs, to achieve the objective (iii) of the work.

The method is designed to work at different spatial scales and with different degrees of detail and information. This structure enables to perform the assessment at national or international scales, for which the ecosystem services association may be unevenly feasible across the area, and thus relying only on the laws and the official documentation provided by the authorities. This is the most basic and flexible level of the analysis, the level 1. When the assessment is focused on smaller scales, e.g. regional or watershed, the assets are further classified with an enriched taxonomy, also including the ecosystem services associated to the defined assets (level 2 of the framework), thus providing a more accurate representation of their value. When instead the assessment aims at describing local flood exposure of environmental assets, e.g. at watershed and municipality scale, a deeper, specific analysis is requested, adding a more detailed, case study specific, list of the ecosystem services associated to the environmental assets in the area (level 3). Level 2 and level 3 are designed to include insights from a participatory based approach. A graphic schematization of the proposed framework is reported in (Figure 1). The framework is incremental, so the assessment always starts with a level 1 analysis, then adding information incrementally for reaching the level 2 or level 3 detail. Step 0 is the collection of the assets in the study area, thus building a dataset of environmental assets, represented in the figure by the blocks with dashed perimeter. The dataset may be enriched and updated while moving through the analysis levels. Step 1 is to determine the listing relevance of the assets, as better described in section 2.2.1, thus creating the updated taxonomy for level 1. After the level 1 weighting procedure (see 2.2.1), the flood hazard information is added to the analysis, thus determining the Environmental Exposure Index (*EEI*) of level 1. Moving to the second level of the analysis, the assessment follows the level 1 taxonomy, which is now enriched with the ecosystem services, thus creating the updated level 2 taxonomy (see section 2.2.2). After the level 2 weighting procedure, the flood hazard information is added and the level 2 *EEI* is obtained. The same workflow applies for level 3 (section 2.2.3).

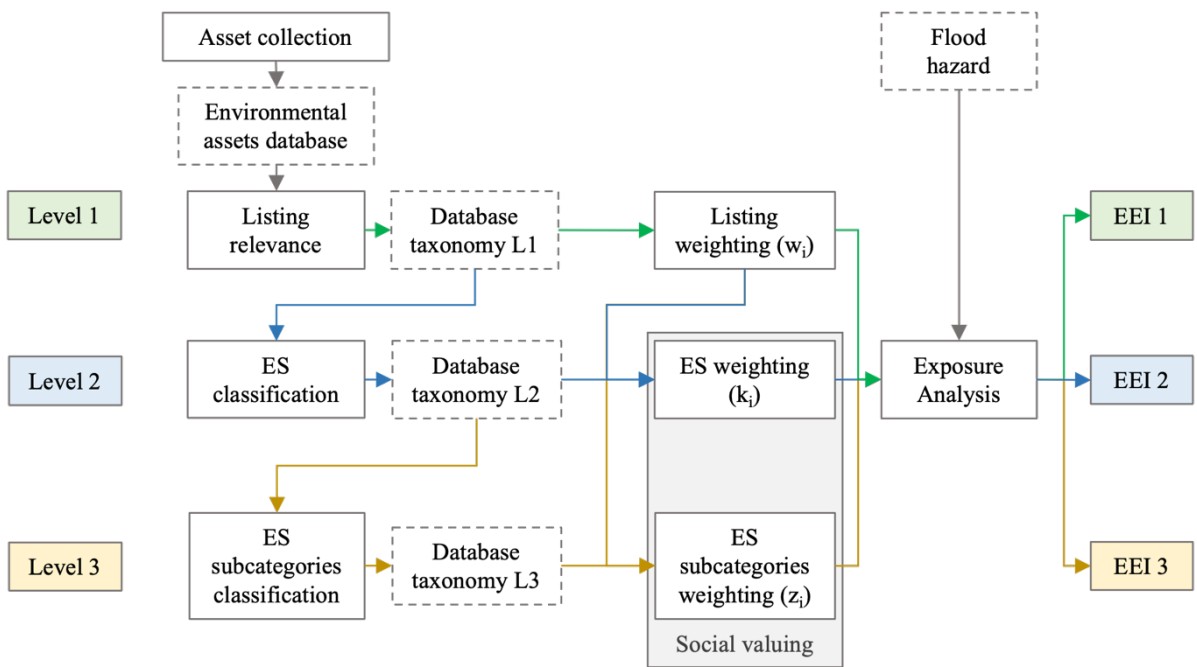

Figure 1. EnvXflood methodological workflow for the determination of the Environmental Exposure Index (*EEI*)
at the three levels of analysis. Ecosystem Services are abbreviated as ES.
In this methodological framework, several variables are defined. The Environmental asset Value $EV_{i,\,l}$ is the
weighted value of the i-th asset in the level of analysis $l$, where $l = \{1,2,3\}$, obtained through a min-max
normalization of the weights. So, $EV_{i,\,l}$ expresses the value attributed to an asset category, given the level of
analysis. The variable $\bar{n}_{i,l}$ is defined for each analysis level and represents the weight assigned to asset $i$.

$$Level\ 1: EV_{i,1} = \frac{n_{i,1} - \min{(n_{i,1})}}{\max(n_{i,1}) - \min(n_{i,1})} \tag{1}$$

$$Level\ 2: EV_{i,2} = \frac{n_{i,2} - \min{(n_{i,2})}}{\max(n_{i,2}) - \min(n_{i,2})} \tag{2}$$

$$Level\ 3: EV_{i,3} = \frac{n_{i,3} - \min{(n_{i,3})}}{\max(n_{i,3}) - \min(n_{i,3})} \tag{3}$$

The description of the weights is reported in sections 2.2.1-2.2.3.
An Equivalence Factor (*EqF*) is defined to determine equivalent units (areas or lengths or numbers, depending on
the asset's geometry type) of the assets, basing on their value $EV_i$, and is obtained by adding a unit to the
environmental asset value $EV_{i,\,l}$. Thus, 1 unit of the most important asset is equivalent to 2 units of the least
important asset, greatly simplifying the understanding of the results obtained by the proposed valuing
methodology. The *EqF* provides a reference asset value (e.g., the least important or the most important), thus
enhancing the interpretation and delivery of the results.
The Environmental asset Exposure Value $EEV_{i,l}$ expresses the exposure of the assets to the flood.

$$EEV_{i,l} = EV_{i,l} \times e_f \tag{4}$$

where $e_f$ is the exposed fraction, i.e., the percentage of exposed area with respect to the total asset area for polygon
features; the percentage of exposed length with respect to the total asset length for line features; the percentage of
exposed number of assets with respect to the total number of assets for point features. When $EEV_{i,l}$ is calculated
on a study area, it highlights the most significant environmental asset exposed, i.e., the most inundated and the
most valuable.
While the above $EV_i$ and $EEV_i$ refer to a single i-th asset category, the overall environmental Exposure Index *EEI*
for the study area, which includes multiple assets categories, is defined as the sum of all the values of the asset
categories, as it follows:

$$EEI_l = \sum_{i=1}^{n} EEV_{i,l} \qquad (5)$$

Where $n$ is the number of the assets considered in the analysis.
The value of the Environmental Exposure Index, *EEI,* represents a flood exposure score which allows making
comparisons among catchments or territories to identify the most exposed areas and assets.
Finally, the ratio between the Environmental Exposure Index and the sum of the values of the assets present in
the area, is defined as Exposed Environmental Fraction, *EEF,* and describes, in percentage, the exposed value
with respect to the maximum total value (*EV*) of the assets in the area. This is an additional indicator, that allows
to rapidly compare the exposure of different study areas and the significance of flood exposure with respect to the
overall environmental assets value of the study area.

$$EEF_l = \frac{EEI_l}{\sum_1^n EV_{i,l}} \qquad (6)$$

The method developed in this study can be applied with different input datasets, but it will produce different
results if the input features are not the same among the analyses. Thus, for each study, it is important to carefully
select the characteristics to be used as descriptors of the assets, being sure that they are uniform and fully
retrievable for all the areas of interest.
It is pointed out that analyses carried out at different levels are not comparable, having different evaluation features
and weights, thus changing the evaluation algorithm.
2.2.1. Level 1
The first level (Eq. 1) is the fastest to be implemented and requires determining the relevance of the assets, based
on the level of listing (local, regional, national, international). International listing includes UNESCO
environmental heritage, but also other assets protected by supranational agreements, such as the Ramsar
convention for the conservation of Wetlands. Level 1 can be easily applied at large scales and thus it can be
suitable for regional/catchment analysis needed in the Flood Risk Management Plans. The spatial database of
Level 1 includes the listing level according to the available information regarding protecting laws/conventions or
recognitions. A weight $w_i$ is assigned to each asset, such that for each step the weight is doubled, starting from 1,
which is for local (i.e., municipal, provincial), then 2 for regional, 4 for national, 8 for international assets
respectively, i.e., $w = \{1,2,4,8\}$. As exemplification, to an asset falling under the UNESCO, Ramsar or
Natura2000 listings, which are international identifications, will be assigned a weight equal to 8, i.e. the maximum
weight. National parks, for instance, are instead usually protected by national laws, and the assigned weight will
be 4. A weight equal to 2 will be assigned to regional parks and all the other assets individuated only by regional
authorities. Some municipalities or provinces will identify some other assets that are relevant only at a local scale.
To these assets, the minimum weight of 1 will be assigned.

2.2.2. Level 2
The second level of analysis (Eq. 2) includes the social value of the environmental asset category, expressed as
the people's perception of the importance of the ecosystem services commonly associated to that asset category.
Among the different ecosystem services classifications, we refer to the one provided by the Millenium Ecosystem
Assessment (MEA, 2005), in which there are four categories: supporting, provisioning, regulating, and cultural.
In the following we refer to these as the "main" ecosystem services categories, and we assigned to them an index
j, j = {1, 2, 3, 4}, such that j=1 is for supporting ES, j=2 is for provisioning, j=3 is for regulating and j=4 is for
cultural ES. For each asset category (e.g., Forests), a review is performed to find existing studies regarding the
ES related to it, thus building a list of ecosystem services associated to each environmental asset category. Where
it was not possible to find specific studies, the analysis was based on expert judgment. In the example of forests,
it is usually recognized that they provide supporting, provisioning, regulating and cultural services. While one
other general example could be the one of the viewpoints, considered as environmental assets, which provide only
cultural ES.
All the information were eventually collected in a spatial database for the Level 2 taxonomy.
For computational simplicity, the information regarding the ecosystem services provided by each asset category
were translated into a matrix $\bar{P}$, $(n \times j)$ with zeroes and ones, with ones meaning that the corresponding ecosystem
service is provided, and zeroes for the opposite.
To distinguish among the j ecosystems services categories introduced above, weights were assigned to them.
Assigning weights to ecosystem services is a common procedure in environmental decision-making, like in Multi-
Criteria Decision Analysis (Adem Esmail and Geneletti, 2018), especially when the goal is to establish a ranking
among those services. Weighting helps resolve trade-offs between conflicting ecosystem services, such as
provisioning (e.g., food production) and regulating services (e.g., carbon sequestration). The significance of
weighting lies in its ability to translate in a simple and effective manner how various ecosystem services are
valued. The column vector $P$, contains the four $p_j$ weights assigned to the ES categories, which can be determined
running a survey, as was done in this study and described in the following section 2.2.4.
Summarizing, the $\bar{p}_{i,j}$ elements of the matrix $\bar{P}$ are, thus, equal to 1 when the j-th ES is attributed to the i-th
environmental asset, 0 when not. Then, multiplying $\bar{P}$, $(n \times j)$ for the ecosystem services weights in the column
vector P, will assign to each environmental asset category their partial weight, the $k_i$. To obtain the final weight
for the Level 2 analysis, $\bar{n}_{i,2}$, the $k_i$ need to be multiplied by the listing level from the Level 1, $w_i$.

$$\bar{P} = \bar{p}_{i,j} = \begin{cases} 1 \implies ES_j \in E_i \\ 0 \implies ES_j \notin E_i \end{cases} \tag{7}$$


$$k_i = \bar{P} \times P \tag{8}$$


$$\bar{n}_{i,2} = k_i \times w_i \tag{9}$$

The $\bar{n}_{i,2}$ are the final weights assigned to each asset category in the Level 2 procedure, which are used in equation
(2) to determine the environmental value $EV_{i2}$, for the Level 2
2.2.3. Level 3
The third level of the analysis (Eq. 3) adds a further classification of environmental assets to create a Level 3
taxonomy and assign the weights $z_i$ (Eq. 10).
For each main category of ecosystem services (supporting, provisioning, regulating, cultural), a sub-set of four
classes of ecosystem services was selected, to be able to catch with more accuracy the properties and the
differences of the assets, and to improve the grip on reality of the analysis. Such classes are representative of the
most common ES for each category, as listed for instance in the Millenium Ecosystem Assessment (MEA, 2005).
They are organized in the array $ES_{sub}$, $(j \times s)$ as shown in figure 2:


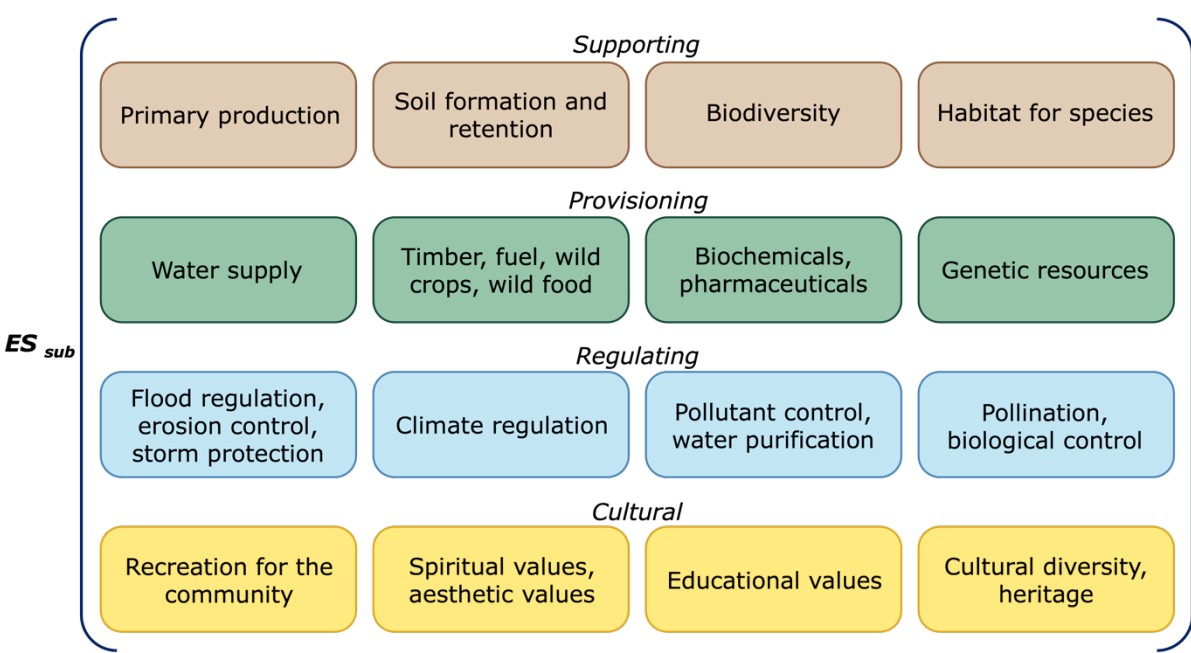

Figure 2: graphical representation of the structure of the ecosystem services subcategories.
For a total of $m = 16$ ecosystem services subcategories.
The index $j$ of the rows represents the corresponding main ES categories, which are the same defined for Level
2. This third level of analysis is intended for the study of smaller areas, due to the high detail of classification
needed. Specific studies or ad-hoc local expert panels can help in defining local environmental assets and in
assigning weights to different ecosystem services sub-categories. In this work the ES subcategory weights $sw_{j,s}$
are assigned based on the survey (sect. 2.2.4) and stored in the matrix $S_w$, ($j$ x $s$), with the same structure of $ES_{sub}$.
It is then defined the matrix $S$, as the product of $P_{diag}$, which stores the weights $p_j$ of the four main ES categories
(the same as Level 2), and the matrix $S_w$ of the ES subcategories weights.

$$S = P_{diag} \times S_w \tag{10}$$

$$P_{diag} = diag(p_j) \tag{11}$$

Similarly to as described for the level 2, the matrix $\bar{S}$, ($n$ x $m$) of zeroes and ones stores 1 if a m-th ES subcategory
is attributed to the i-th asset and allows to apply the ES subcategory weights selectively to only the assets which
provide those ES. Thus, the elements $\bar{s}_{i,m}$ of the matrix $\bar{S}$ are equal to 1 when the m-th ES subcategory is attributed
to the i-th environmental asset, otherwise are 0

$$\bar{S} = \bar{s}_{i,m} = \begin{cases} 1 \implies ES_{sub_m} \in E_i \\ 0 \implies ES_{sub_m} \notin E_i \end{cases} \tag{12}$$

Eventually, the partial $z_i$ (Eq. 10) weights are assigned to each asset, and they can then be used in the Eq. (3).

$$z_i = \bar{S} \times S_c \tag{13}$$

Here, the column vector $S_c$, ($m$ x $1$) is obtained by arranging in a single column the elements of $S$, row by row.

$$\bar{n}_{i,3} = k_i \times w_i \times z_i \tag{14}$$

Eventually, the $\bar{n}_{i,3}$ in the equation (14), represents the weight of an asset in the Level 3 analysis, and it is used to
determine the environmental value in the $EV_{i3}$ in equation (3).
2.3. The survey

The survey was developed by means of the Google Forms web platform (supplementary material), targeting a
group of individuals familiar with environmental and flood-related topics, though not necessarily experts in
ecosystem services or environmental assets. The targeting choice is based on the rational of acquiring insights
from people able to fully understand the proposed questions, but without limiting the audience only to
environmental experts. Different and multiple targeting is possible, and the results may be eventually aggregated
in one. This participatory approach follows a basic but effective version of methodologies commonly used in
multi-criteria decision making/analysis (MCDM/A), already proven to be meaningful and suitable for flood risk
assessment (Evers et al., 2018; Hansson et al., 2013) and, more broadly in similar sectors (Ferla et al., 2024),
where stakeholder input is essential for capturing complex and broad-ranging relationships, here with the objective
of determining priority in the environmental management and protection. The survey asks to rank the ES category
(for the Level 2 classification) and sub-categories (for the Level 3 classification) from the most to the least
important. The weights $w = \{1, 2, 3, 4\}$ are assigned as the following: the highest weight, 4, goes to the first
classified, and the lower weight, 1, goes to the last. To catch the degree of consensus among respondents, a decimal
value representing the proportion of responses ($s = share$) that selected each category was appended to the
assigned weight. This approach retains information about the share of participants who selected each option,
providing insight into the uncertainty or variation in public opinion regarding the importance of each category.
For exemplification, following the equation 15, if a category has been voted as the second most important [$2^{nd}$ =
weight 3] by the 50% of the respondents [share = 0,50], its $sw_{j,s}$ weight for the matrix $S_w (j \times s)$ in equation 10
would be 3,5.

$$sw_{j,s} = w + s \qquad\qquad (15)$$


Where $w$ are the raw weights derived from the pure ranking, and $s$ is the share of the responses, as described
above.

2.4. Case studies: Tuscany - Italy

The study area for applying levels 1 and 2 of the analysis is the Tuscany region, in central Italy (Figure 3, panel
A, B). Tuscany extends for about 23000 km$^2$ and its morphology includes mountain chains and some plains, but
it is dominated by hills, which occupy approximately 66% of the area. Its main river is the Arno River, which has
a length of about 241 km, and a catchment area of about 8288 Km$^2$.
Only the portion of the regional area managed by the Northern Apennines River Basin District Authority, which
covers approximately the whole region, is comprised in the present study.
For the analysis of level 3, two catchments in the Region are selected to compare the results: the Orcia and the
Chiana valleys (Figure 3, panel C).
The Orcia Valley is in the south-east of the Tuscany region and took its name from the Orcia River, which has a
length of about 57 km, flows from East to West, and has an overall watershed surface area of about 798 km$^2$,
considering the basin delineation named "S. Angelo Cinigiano" in the dataset provided by the Tuscany regional
authority for hydrology (SIR). A portion of the valley has been inscribed in the UNESCO World Heritage Sites
for its landscape's distinctive aesthetics, since 2004.
The Chiana Valley is morphologically flatter than the Orcia Valley, its main drainage canal is the "Canale Maestro
della Chiana", which is a 62 km length artificial channel flowing from South to North. The watershed surface area
is about 1290 km$^2$. Many attempts of reclamation were made in the past since ancient times, and they eventually
resulted in the completion of the "Canale Maestro della Chiana" and its network of tributaries. The channel starts
near Chiusi Lake, and it is a left tributary of the Arno River. The confluence is located near the city of Arezzo.
The Chiana Valley watershed area studied here is a sub-basin of the Arno River basin, identified by the name
"Ponte Ferrovia FI-Roma" in the basin delineation provided by the Tuscany regional authority for hydrology
(SIR).
The list of environmental assets included in the spatial database for the whole Tuscany and for the Orcia and
Chiana Valley is available as supplementary material, and all the information has been retrieved from public
datasets of the official authorities at regional, national and international level.

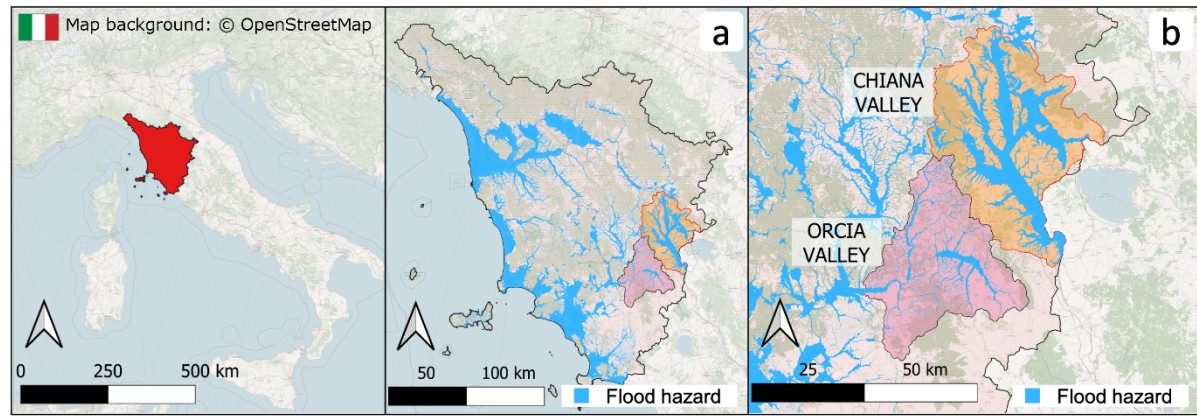

Figure 3. Case studies identification. Tuscany region for Levels 1, 2 (a); Chiana and Orcia valleys for Level 3 (b). Flood hazard areas are depicted in blue (flood hazard extent: Autorità di bacino distrettuale dell'Appennino Settentrionale). Map background: © OpenStreetMap contributors 2023. Distributed under the Open Data Commons Open Database License (ODbL) v1.0.

2.5. Flood hazard

The hazard assessment was carried out with the official flood hazard maps made available according to the European directives 2000/60/CE and 2007/60/CE, provided by "Autorità di bacino distrettuale dell'Appennino Settentrionale", within the Flood Risk Management Plan (FRMP), (PGRA – Piano Gestione Rischio Alluvioni). The maps were employed in the study to assess the flood extent and thus the areas directly exposed to the flood hazard. The maps refer to three hazard levels, P1 is the low, P2 is the medium and P3 is the high hazard level. The analysis was based on the low probability hazard scenario P1.

3. Results and Discussions

3.1 Environmental assets taxonomy

The following diagram, (Figure 4) summarizes the environmental assets considered and collected to create the baseline geospatial database. The proposed taxonomy, as already introduced, has been initially defined taking advantage from the most relevant international laws for environmental assets conservation and protection. It is divided in 4 macro categories, embracing all the collected assets. They are:

- Water resources and ecosystems.
- Geologic sites.
- Terrestrial ecosystems.
- Landscapes.

Intermediate categories have been defined for each macro class, providing a more transferable taxonomy, which include freshwater bodies, coastal areas and transitional waters, landforms, underground geosites, fossil bearing layers, wildlife sanctuaries, parks, terrestrial habitats, land scenery, sightseeing spots or trails. The last branches of the scheme are populated by the specific environmental assets that we were able to identify. While moving among different areas, the onomastics may vary, and some adaptation may be necessary, though most of the assets can be represented or included in the proposed list.

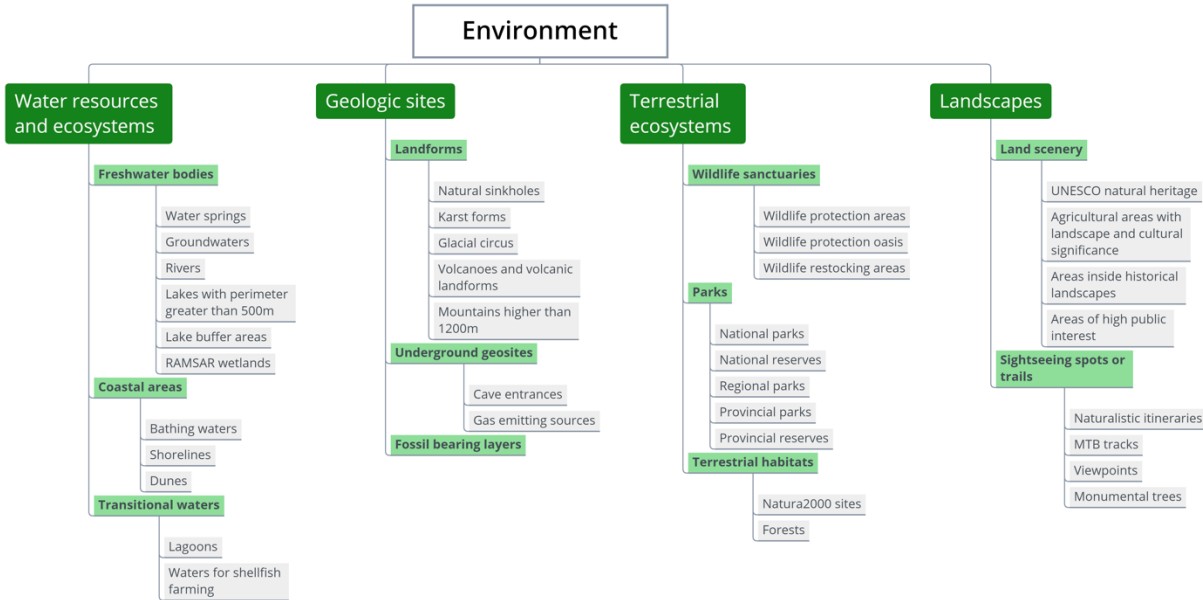

Figure 4. Taxonomy of the most relevant environmental assets, categorized into i) Water resources and
ecosystems; ii) Geologic sites; iii) Terrestrial ecosystems; iv) Landscapes.
Water bodies, wetlands (e.g., RAMSAR areas), rivers, and lakes are explicitly considered in the flood exposure
analysis carried out in this work, highlighting their relevant involvement in floods. Despite this, they are usually
excluded from common flood impact and risk analyses as water bodies themselves, adopting too strong
simplifications, which are retained to be no more adequate to correctly represent the phenomenon.
3.2. Survey results
The survey received about 65 answers. 63% of them were provided by students, researchers and professionals in
the field of water and environmental sciences and engineering.
The following table (tab. 1) reports the weights to be used in the level 2 and 3 analyses, resulting from the
processing of the survey's answers.
Table 1: Weights applied to the ES categories, resulting from the survey. At level 2, the main ES categories are
shown. At level 3, the respective sub-categories are reported.

| Level 2 | | Level 3 | |
|---|---|---|---|
| ES main category | ES main category weights: $p_j$ | ES sub-category | ES subcategory weights: $sw_{j,s}$ |
| Supporting | 4,33 | Biodiversity | 4,33 |
| | | Primary production | 3,31 |
| | | Soil formation | 2,33 |
| | | Habitat | 1,33 |
| Regulating | 3,30 | Climate regulation | 4,50 |
| | | Pollutant control | 3,42 |
| | | Flood, erosion control | 2,30 |
| | | Biological control | 1,34 |
| Provisioning | 2,28 | Water | 4,88 |
| | | Timber, fuel, … | 3,42 |
| | | Biochemicals | 2,39 |
| | | Genetic resources | 1,39 |
| Cultural | 1,61 | Educational | 4,45 |
| | | Cultural heritage | 3,45 |
| | | Recreation | 2,34 |
| | | Spiritual values | 1,45 |


The Supporting ES category turned out to be the most important. Among its ES subcategories, Biodiversity is
placed first, followed by Primary production, Soil formation, and Habitat. The share of the answers, expressed by
the decimals of the weights, was around 30% for all the choices, indicating a homogeneous distribution of the
answers. The Regulating ES category resulted to be the second most important ES main category. Among its ES
subcategories, Climate regulation was voted as the most important, with a good degree of accordance (50%). The
Provisioning ES placed third among the main ES, and the Water subcategory was voted the first, with a high
degree of accordance (88%). The last main ES was the Cultural one, with 61% of accordance, and the most
important subcategory was the Educational one.
Due to the characteristics of the topic, it is considered appropriate to potentially open the survey to a wider range
of expertise, including, for example, biologists, economists and cultural heritage experts. Local and regional
stakeholders could furthermore be involved, aiming at reaching a better policy impact and making the analysis
the most fitted possible to the study area. The selected weights should be the most shared possible; though, they
remain related to the social, historical, and *environmental* context and time in which the assets are evaluated and
are strictly dependent on the scale of the project. It's relevant to point out that the framework of the EnvXflood
method can also work with different sets of weights, and it is also possible to perform parallel analyses of the
same areas, applying different weights. This allows to compare the environmental assets' exposure to floods, for
instance, from two or more different points of view, such as the ones of different stakeholders, creating seminal
comparative results for the decision-making processes and the authorities.

3.3. Tuscany region results

The methodology, as already discussed, was designed to work with three levels of analysis. The different insights
obtained through the three levels make it possible to perform very rapid (level 1), still meaningful, analyses in
case of post-disaster assessments of assets hit by a flood, as well as very detailed evaluations (level 2, level 3),
more suitable to prevention and planning measures, thus making this framework adaptable to multiple necessities
and different scenarios. The second level of analysis is well-balanced among resources (time, data) and results
obtained and it could be effectively applied at regional scales. The third level requires carrying out site-specific
studies during all the phases of the analysis, implying a considerable amount of time and resources. It is more
suitable for applications at small scales, like protected areas, and sub-basins (e.g., valleys).
In this study, the method developed was applied to the Tuscany region, in Italy. The level 1 and level 2 analyses
were performed for the whole region. Figure 5 reports the most significant results of the second-level analysis.
The figure is composed of a map on the left, and a diagram on the right, which also represents the legend for the
colour ramp adopted in the map. The environmental asset flood Exposure Value $EEV_{i,2}$, is plotted on the top axis
of the diagram, and it is graphically represented by the grading-coloured line (from red: most exposed; to green:
less exposed). Plotted on the bottom axis of the chart is also reported the equivalence factor $EqF$, graphically
represented in the diagram by the grey vertical segments. This set of information already provides a complete
view of the analysis of the assets, expressing how much the assets are significant ($EqF$), and the weighing scale
between their value and their physical exposure to the hazard ($EEV_i$), i.e., the flood.
The overall Environmental Exposure Index $EEI_2$, and the Exposed Environmental Fraction $EEF_2$, are reported in
Table 2. The equivalence factor $EqF_i$, and the Exposed Environmental Value, $EEV_i$, are designed for a comparison
among the assets within the study area, while the $EEI_2$ and the $EEF_2$ are intended for a comparison among different,
but similar areas, as far as they are homogeneous in the data availability. The total Environmental Value $EV_2$
obtained in the analysis is also reported on the map.

Table 2: Resulting indicators of the Level 2 analysis carried out for the Tuscany region.

| Level 2 analysis | $EEI_2$ | $EEF_2$ | $EV_2$ |
|---|---|---|---|
| Tuscany | 4,7 | 33 % | 14,1 |


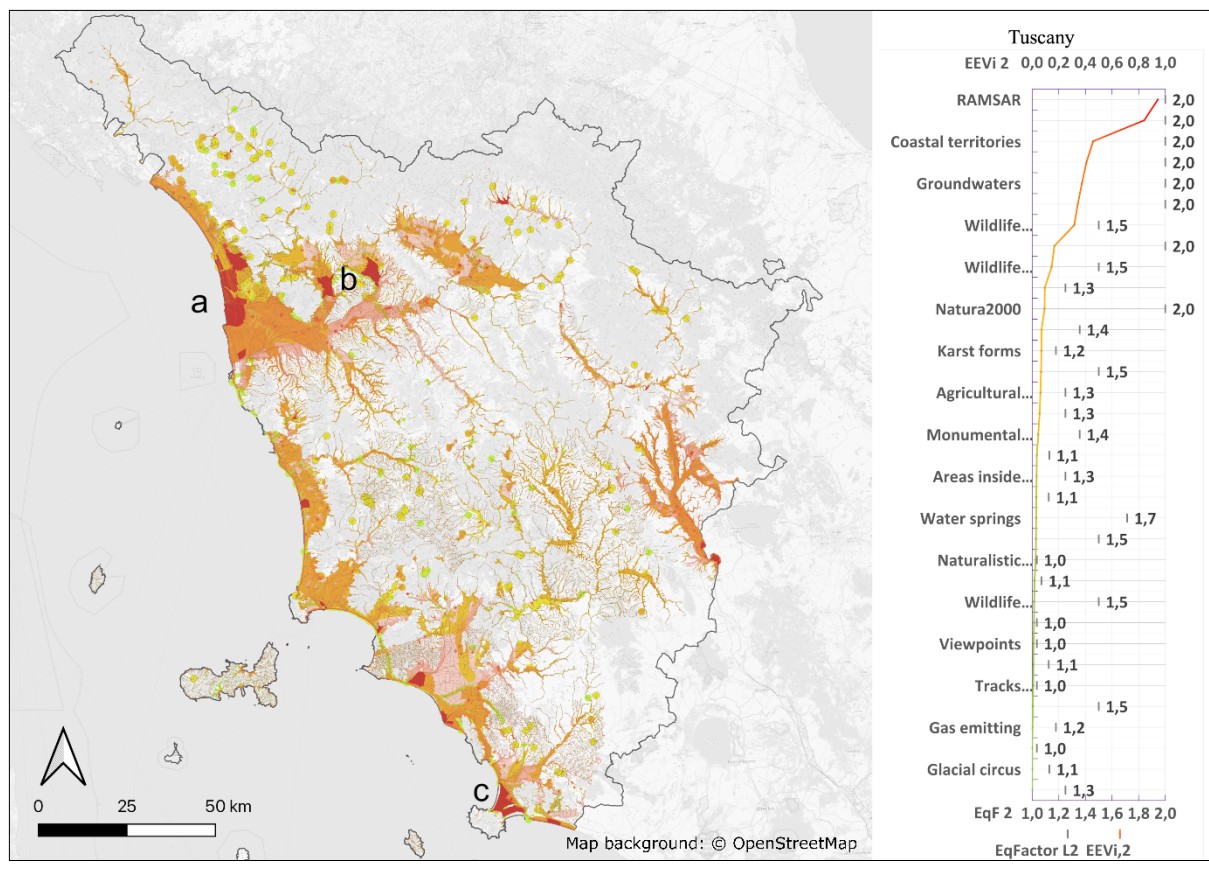

Figure 5. Flood exposure of the environmental assets of the Tuscany region, the most exposed environmental assets are shown in red, progressively grading to yellow and green, depending on their ranking in the Level 2 analysis. The areas with high exposure values marked with a, b, and c represent Massaciuccoli Lake, Fucecchio swamps, and Orbetello Lagoon, respectively. Map background: © OpenStreetMap contributors 2023. Distributed under the Open Data Commons Open Database License (ODbL) v1.0.

The *EEF* indicator provides a direct and very effective reading of the flood exposure of the assets of the region, which, for the Tuscany region, is about 33%. The *EEF* is a large-scale indicator, useful for comparisons among different areas, but to detail the knowledge of the flood exposure of the assets in the area, it is necessary to focus on the Environmental Exposure Value $EEV_i$ of each asset. Water-related assets, are, as expected, at the first places of the rank. This means that they are the most valuable assets and the most flooded assets too. This result must not be taken for granted, and it is strongly believed that it is necessary to include water-related assets in the flood risk assessments, since often they are not. Assessing their exposure to floods brings important information in the knowledge of the territory and of the hazard, allowing better responses in case of necessity (pollution spread, physical damages, habitats or ecosystems losses, …).

The most exposed assets are the RAMSAR areas, followed by the lakes (coloured in red in Figure 5, as the Massaciuccoli lake -highlighted by "a"-, the Fucecchio swamps - highlighted by "b"- and the Orbetello Lagoon - highlighted by "c"-), the coastal territories, and the lake buffer areas (in dark orange in Figure 5). Groundwaters (in this study considered as the "footprint" of the aquifer recharge) and rivers are in the fifth and sixth position respectively. From this point on, the two rankings (level 1, level 2) become distinct, because the differences in the EV computed in the two analyses are more pronounced. In level 1, not reported here, the EV is only guided by the level of protection, i.e., legislative listing. Instead in level 2 also the ES provided by the assets are included, to describe their importance at an ecosystem, environmental and social level, thus providing a different, more significant, ranking. A good exemplification could be the one of the MTB Tracks: they are listed at the regional level, thus ranking 14th/34 in the level 1 analysis. In level 2, they are recognized to provide only a few ES (cultural), thus, despite the regional listing, they fall to the end of the ranking, leaving the higher places to the most important assets (assets providing more Ecosystem Services).

From a scientific and engineering point of view, to know which assets are more exposed to floods than others, in a way able to catch the role of the assets in the ecosystem and in the society, therefore getting a measure of their value, is a great step forward. This result opens new perspectives in the management of flood risk. Firstly, aligning the environmental exposure analyses outcomes to the common exposure definition used in risk analyses, such as

buildings' exposure, makes it possible to integrate the environmental assets' exposure into conventional risk
equations. Furthermore, using Ecosystem Services as part of the evaluation guarantees approaching the theme in
a holistic manner, not focusing only on a single sight of it. Secondly, this mode of assessing flood exposure
consents to better move to the next research phases (e.g. vulnerability assessments), straightforwardly prioritizing
the most exposed assets, and creating the conditions for rapid growth in research and significant improvements in
flood risk assessments for environmental assets. Advancements should then focus on the environmental assets'
vulnerability to floods, explicitly considering the peculiarities of floods in the Anthropocene.
Back to the map, reporting the Equivalence Factor along with the $EEV$ has the aim of stressing the social,
environmental, and, indirectly, also economic values expressed through the ES provided by the assets, which are
included in the $EEV$. The most valuable assets have the highest $EqF$, and most of them are in first places.
Nevertheless, other valuable assets, like the Natura2000 and the UNESCO assets are not as much exposed as
RAMSAR or lakes assets, thus positioning lower in the $EEV$ ranking, because they are less flooded. This
exemplifies well how the model is capable to rank efficiently the assets keeping all the important aspects in the
computations. The areal extension of the environmental assets exposed to floods in the Tuscany region is clearly
reported in Figure 4. In the map it is also observable the exposure extension of the coasts and the coastal territories
of Tuscany, which are almost completely highly exposed to floods.
3.3.1. Orcia Valley and Chiana Valley results
For the Orcia and the Chiana valleys, the analysis was pushed to the third level, thus including more details about
the ecosystem services provided by the assets. The following figures (Figure 6, Figure 7) report the main
outcomes. The figures are composed of the same elements described in the previous section. The Environmental
asset Exposure Value $EEV_{i,3}$, is plotted on the top axis of the diagram, and it is graphically represented by the
grading-coloured line (from red: most exposed; to green: less exposed). Plotted on the bottom axis of the chart is
also reported the equivalence factor $EqF$, graphically represented in the diagram by the grey vertical segments.
The overall environmental Exposure Index $EEI_3$, the Exposed Environmental Fraction $EEF_3$, and the
Environmental Value $EV_3$, are reported in Table 3.
Table 3: resulting indicators of the Level 3 analysis carried out for the Orcia and Chiana valleys.

| Level 3 analysis | $EEI_3$ | $EEF_3$ | $EV_3$ |
| --- | --- | --- | --- |
| Orcia Valley | 1,8 | 25 % | 7,28 |
| Chiana Valley | 3,0 | 51 % | 5,94 |

The results of the Level 3 analysis performed for the Orcia and the Chiana valleys are fully comparable. These
outcomes can be used by the regional authority to prioritize further studies, focusing on assessing the flood
vulnerability of the most exposed assets and areas, eventually planning mitigation measures where they are most
necessary, effectively minimizing the environmental and social losses. It is evident, from analysis outcomes that
the environmental assets of the Chiana Valley are more exposed to floods than those in the Orcia Valley. The
Chiana Valley is morphologically flatter than the Orcia Valley, and it presents also other characteristics which
favour flooding. It also has several lakes and wet areas, as highlighted in red in Figure 7 and the drainage network
is largely artificial. Two major lakes are located to the south, the Chiusi Lake (Figure 7, a) and the Montepulciano
Lake, which is also a natural reserve (Figure 7, b). Instead, the Orcia Valley has a very dense drainage network
(Figure 6), and only a few lakes. The analysis pointed out that the environmental value $EV$ of the Orcia Valley is
greater than the Chiana Valley (Table 3) since, for instance, UNESCO assets are not present in the Chiana Valley,
as for the monumental trees, karst springs, and cave entrances. However, the Environmental Exposure fraction
$EEF$ of the Chiana Valley is approximately double of the Orcia Valley, and the same is for the $EEI$ index, due to
greater flood extension. Thus, even if the value of the assets is lower, the indicators show that the environmental
assets' exposure to floods is higher in the Chiana Valley. The $EqF$ values become particularly effective in this
comparison, highlighting those significative assets which are not largely flooded, but deserve more attention in
the analyses due to their environmental value. This is the case of UNESCO and Natura2000 assets in Orcia Valley.
The $EqF$ can be a guide for further, asset-specific analyses, to better assess the exposure and, eventually, the flood
risk of the most important assets.
Overall, rivers are the most exposed assets in the Orcia Valley, followed by the lakes and their buffer areas, water
and karst springs. Regarding the Chiana Valley, the most exposed assets result to be the lakes, their buffer areas,
the rivers, the Natura2000 areas, and the groundwaters. The Chiana Valley lakes have almost double the exposure
value than in the Orcia Valley. Even if at the third position, the rivers have a higher exposure value (proportionally)
in the Chiana Valley than in the Orcia Valley, due to the reasons discussed above.
Natura2000 assets are present in both valleys, and they are more exposed in the Chiana Valley.

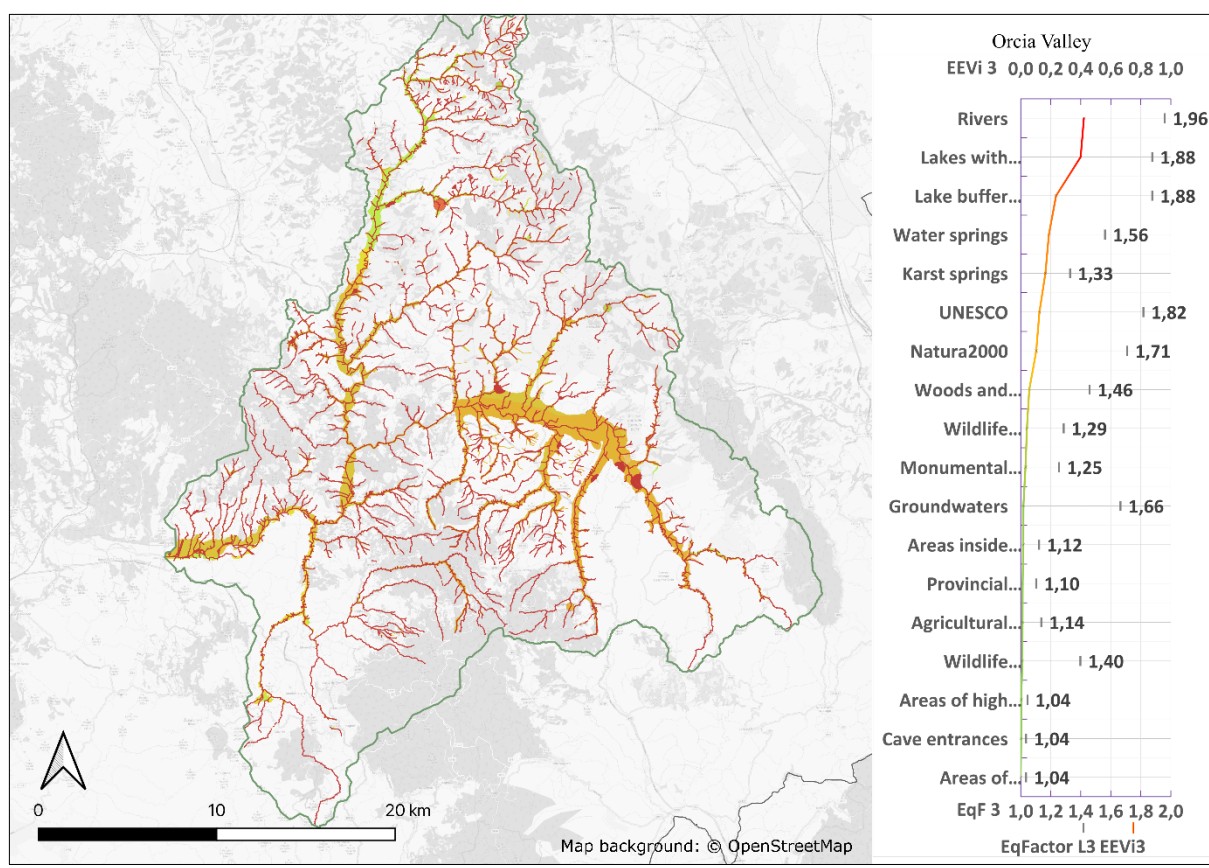

Figure 6. Flood exposure of the environmental assets of the Orcia Valley The most exposed environmental assets
are in red, progressively grading to yellow and green, depending on their ranking from the Level 3 analysis. Map
background: © OpenStreetMap contributors 2023. Distributed under the Open Data Commons Open Database
License (ODbL) v1.0.

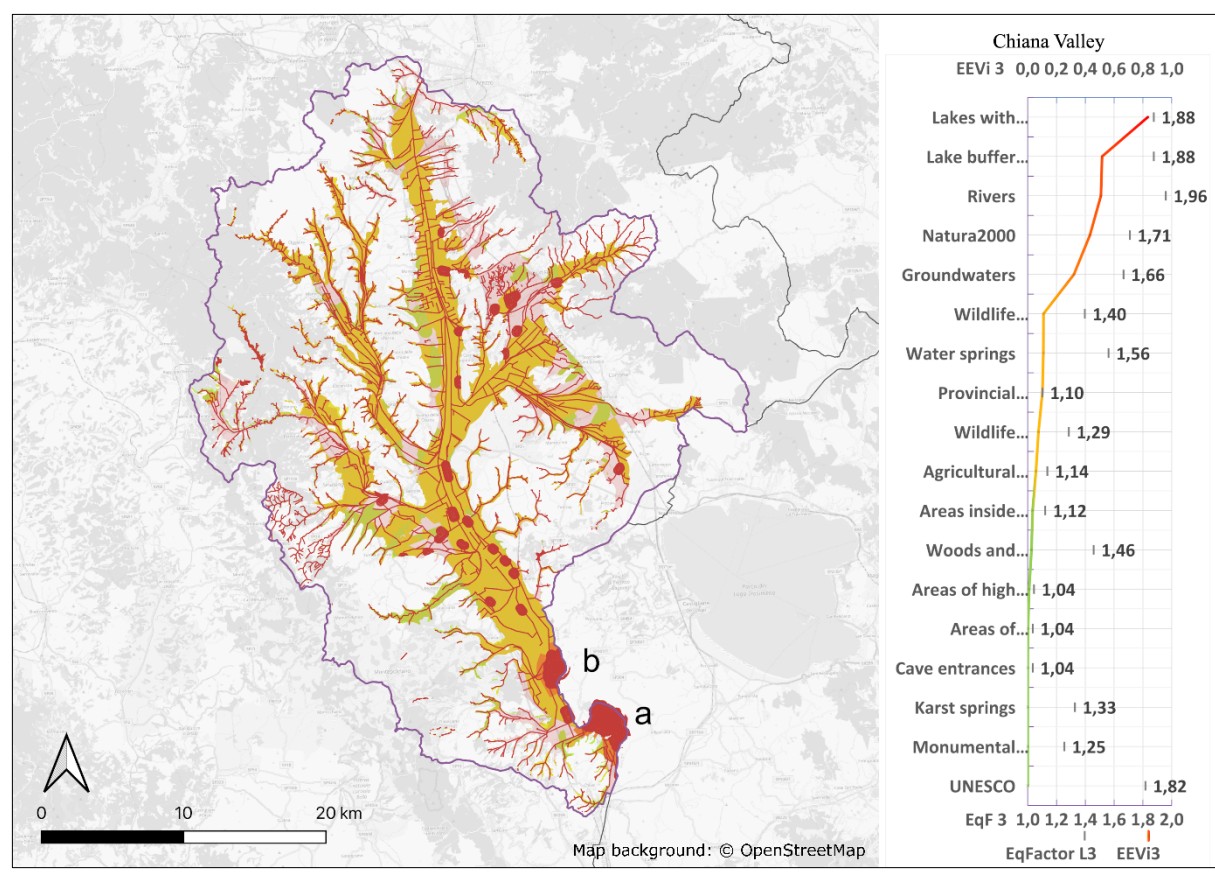

Figure 7. Flood exposure of the environmental assets of the Chiana Valley. The most exposed environmental assets are in red, progressively grading to yellow and green, depending on their ranking from the Level 3 analysis. In the map are highlighted the Chiusi Lake (a) and the Natural Reserve of the Montepulciano Lake (b). Map background: © OpenStreetMap contributors 2023. Distributed under the Open Data Commons Open Database License (ODbL) v1.0.

## 4. Discussion and conclusions

Flood risk assessment of environmental assets is a process that currently lacks its fundamentals, such as shared and effective definitions and methodologies to assess their exposure and vulnerability to flooding. This study aimed at providing an environmental asset taxonomy (research objective (i)), which has been defined taking advantage from the most relevant international laws for environmental assets conservation and protection. The proposed taxonomy was then integrated with more detailed environmental assets categories, defined among the ones already present in the European and Italian legislative framework, adapted with intermediate categories to enhance its transferability, without limiting its application to the case study examined in the present work. This taxonomy can help researchers and practitioners to properly recognize environmental assets to be comprised in flood risk analyses and can be adapted to fit local peculiarities if required. The four main categories, i.e., Water resources and Ecosystems, Geologic sites, Terrestrial Ecosystems, and Landscapes, are wide-ranging and easy to apply also in different settings, without needing further adaptations. The second step of the study was the development of a method, named EnvXflood, to estimate flood exposure of environmental assets (research objective (ii), delivering the overall Environmental Exposure Index ($EEI$) (research objective (iii)). Exposure assessment focuses on the social and environmental value of the assets, beyond the flooded area analysis, also through the evaluation of the Ecosystem Services provided by each environmental asset category. Social values were investigated by means of a participatory approach. The methodology developed in this study is structured across three levels of detail requiring increasing information, from fast analyses suitable for regional assessment (Level 1 and Level 2) to a detailed ecosystem-service-based site analysis (Level 3). The method outcome is the ranking of the environmental assets, ordered from the most important and most flooded to the least important and less flooded. The application of the method to the study area in Italy (Tuscany region, Chiana, and Orcia basins) highlighted that the environmental assets related to water, such as rivers, lakes, and wetlands, are the assets most exposed to floods, and among the most valuable in terms of ecosystem services provided. Despite this, water bodies are often neglected in flood risk analysis, assuming that floodings are not damaging natural areas, thus not requiring a sound and comprehensive flood risk analysis. This assumption is no more considered acceptable since the human activity deeply changed natural areas, and many aspects are emerging from the studies on potential impacts (Arrighi and Domeneghetti, 2024). During and after a flood, ecosystem services delivery is altered and may be disrupted for a certain time (Dodd et al., 2023), the habitat provisioning service may be interrupted (Ciampittiello et al., 2022), pollutant may be transported with effects on ecosystems and to health (Weber et al., 2023). Extreme floods can alter significantly the aquatic ecosystems and the ecosystem services they provide (Talbot et al., 2018).

Moreover, flood impacts have been assessed also on the biodiversity of terrestrial animals, with the severity depending on various factors such as flood duration, and depth (Zhang et al., 2021), but due to the anthropogenic alterations, also affecting the biodiversity in riverine systems (Walker et al., 2022). Also, floods significantly impact lake ecosystems by altering their hydrological characteristics, affecting water quality, salinity, and biological processes (Muduli et al., 2022). Further research should aim at consolidating the asset taxonomy for flood exposure analysis and their social value, moving towards a consistent understanding of environmental flood impacts. Moreover, a standardized procedure for the weighting process, and standardized databases of the environmental assets, officially made available by authorities, would represent improvements effectively fostering comparison among regions, also if they are controlled by different administrations. This work was developed to be the first step forward towards a better, more informed, and more comparable, flood exposure assessment of environmental assets, and so, to a better flood risk assessment. Scientific community and authorities working at any spatial scale, strongly need commonly accepted procedures and shared knowledge to improve the research on, and the management of, environmental assets, and the outcomes of this work aim at filling this current gap. Indeed, as it is a novel approach in a field not well documented by the literature, it includes some uncertainties, especially regarding the weight selection. While the individuation of the environmental assets categories relies on laws and official datasets, the weights are representing the opinion of the interviewed people regarding the importance of the Ecosystem Services associated to the assets. The results reflect the diverse social, economic, educational, and professional backgrounds of the respondents, as well as their personal experiences and the local context in which they reside. Despite this diversity, the derived weights are still considered robust and accurately representing the relative importance of Ecosystem Services (ES) and their roles, in line with the structured participatory approach based on Multi-Criteria Decision-Making/Analysis (MCDM/A) methodologies (e.g., (Evers et al., 2018; Ferla et al., 2024; Hansson et al., 2013)). While future surveys or expert consultations could provide further refinements, especially if applied to areas in which social context deeply different from the one of our audience, significant variations in the current findings are not anticipated. Slights variations are expected also changing the professional background of the audience, as well as if moving to the industry sector or to a wider, generalized and less informed public, e.g. residents. Nevertheless, additional participatory approach experts' validation is recommended to enhance the robustness and reliability of the results.

Other source of uncertainty is the partial subjectivity included in the attribution of the ecosystem services to the
environmental assets, which, wherever possible, was conducted referring to the literature, with some expert
opinion integration when necessary.
Data availability
GIS data will be made available in a public repository after acceptance.
Author contributions
All authors contributed to the study conception and design. Material preparation, data collection and analysis were
performed by Gabriele Bertoli and Chiara Arrighi. The first draft of the manuscript was written by Gabriele Bertoli
and Chiara Arrighi and all authors commented on previous versions of the manuscript. All authors read and
approved the final manuscript.
Competing interest
The authors declare that they have no conflict of interest.
Acknowledgments

This study was carried out within the RETURN Extended Partnership and received funding from the European
Union Next-GenerationEU (National Recovery and Resilience Plan – NRRP, Mission 4, Component 2,
Investment 1.3 – D.D. 1243 2/8/2022, PE0000005)

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

830 --

