# Peer review of "Flood exposure of environmental assets"

_Natural Hazards and Earth System Sciences, 2024_

## Author Response (AR1)

**Responses to reviewers**

Please, be aware that:
- all the line numbers cited in the replies refer to the document with the track of changes.
- the list of the references at the end of the manuscript has been updated only in the document without the track of changes, to prevent issues within the changes tracking and the reference manager.

**Reviewer 1 (19/07/2024)**

**Comment 1**

Several components within the text can be introduced better. It would be helpful to get the introduction and definition of environmental assets earlier on in the text, rather than in section 2.1. Moreover, the focus on Italy is brought in quite abruptly in line 55, without explicit introduction of the case study. Similarly, the introduction of the EnvXflood method can be improved (in line 150), by including one or two additional lines on what it is, incl. at least one source. The model is also not mentioned in the introduction, only previously in the abstract.

**Reply**

We agree that the concept of Environmental Assets could be introduced earlier in the text, to clarify the definition since the beginning. We rearranged the text following your suggestions, modifying it in lines 34 - 36. Italian case study mention moved at the end of the intro at line 133. On the other side, EnvXflood is the method that we developed in this work, thus there aren't external sources to mention.

**Comment 2**

In section 2 it takes a long time before it can be understood what the authors have done, because of the way that the chapter is structured. E.g., in line 137-130 "they are usually protected by national or regional laws, which can be used as identification instruments. After identifying the assets commonly protected at the European level (and the Italian level) a classification based on few typologies has been proposed as a taxonomy for environmental assets". In these few lines, it's assumed after the first sentence that the reader understands that it's not only possible to identify assets using laws, but that this is indeed what the authors have done (while a full explanation of this is not provided until chapter 2.21 in line 219). Also, it is assumed that the reader understands that the chosen case study is Italy, while this is not explicitly mentioned anywhere in the text until section 2.4 in line 317. Moreover, in line 160, different scales are mentioned and the authors refer to Figure 1, however at that point in the text it is not yet clear if the authors refer to spatial or temporal scales and how the different levels in the figure refer to different complexities. I suggest refraining from referring to the figure until the figure can be understood by the reader or adding additional information to the figure to clarify.

**Reply**

Thanks for pointing out the issues related to clarity and organization of the text. We see that adding some more detailed explanations and relocating some sentences will facilitate the

Gabriele Bertoli, Chiara Arrighi, Enrica Caporali

comprehension of the writing. We intervened in the text from line 159 on, until line 171. We avoided to refer to the Italian case study here, as correctly pointed out. The second part of the comment, regarding the scale and the low clarity deriving from referring to the figure too early, have been addressed by moving the description before the figure, and adding a comprehensive explanation of the figure, accompanying the reader through each block of the diagram. Changes are in place from line 195 to 228.

**Comment 3**

There is quite a strong focus on the importance of water resources and ecosystems and while it's understood that they show up as important in the results, the authors already emphasize their importance specifically before discussing the results (lines 376-385). They are the only typology of which the vulnerabilities are discussed, while for example landslides in land-based ecosystems can also cause cascading risks.

**Reply**

We gave a lot of importance to the water related assets because we observed a major issue in the common practice: despite their primary involvement in floods, and despite their importance, they are commonly not included in flood risk analyses, and we really want to emphasize that this practice must change. However, we totally agree that they are not the only typology that can suffer from flood events and that the more comprehensive these analyses are, the more we are able to catch all the exposed and – potentially – vulnerable elements. To better clarify, the introductory sentences have been rearranged clarifying that we focus on river floods, even if they are not the only hazard, and adding examples of a variety of expected impacts on the ecosystems and on the environment from line 36 to line 60, also adding more recent and varied references.

**Comment 4 – minor adjustments –**

Some minor issues with the abbreviations used: ES is already used in line 155 before introducing the abbreviation in line 172. Also, in the caption of figure 1 in line 165 it says "ES stays for Ecosystem Services" which seems to be a translation/textual error, perhaps replace with "Ecosystem Services is abbreviated as ES" or something similar. It would also be good to check for consistency regarding the use of italics in abbreviations, e.g., in line 158 EEI is in italics, while in 164 EEI is not in italics.
Tait 2019 in line 110 is not included in reference list, also how the source is referenced now it seems to support the statement made in the sentence, while, upon further inspection, it seems to be an example of one of these rare studies mentioned in the sentence. Please clarify that this is an example only.

**Reply**

Regarding the minor adjustments, thanks again for the attention you paid in examining our manuscript. We solved the ES abbreviation issue, and the caption of Figure 1 was updated as suggested.
We also fixed the use of italic style in the manuscript.
The Tait example has been clarified by moving the reference in the correct portion of the sentence, and it has correctly been added to the reference list.

**Reviewer 2 (10/08/2024)**

**Comments to "Introduction" section**

**Comment 1**

The choice of the term "environmental asset" could be supported by a reference.

**Reply**

We agree that the term "environmental asset", even if of common use, could benefit from additional support. We added a better description, and we incorporated references to substantiate this terminology. Both in the description (lines 34 to 35) and, more detailed, in the section 2.1, from line 151 to 155.

**Comments 2, 3**

The risk definition could be updated, considering that risk and the associated damage are a result of all three components hazard, exposure and vulnerability. Currently, the article mentions "the expected damage for the given hazard".
The authors should update the risk definition. I would recommend (1) replacing the reference ("mod. From UNDRR") with a concrete reference that is shown in the bibliography and (2) updating the following aspects: (a) "object of the risk analysis" could be replaced by "elements at risk" and (b) the term vulnerability should be described. For alignment with the UNDRR definition (vulnerability = "conditions [...] which increase the susceptibility of an individual, a community, assets or systems to the impacts of hazards"), the section describing "vulnerability V, or the expected damage for the given hazard") should be revised.

**Reply**

We acknowledge the importance of aligning our definition of risk with the established framework, considering the components of hazard, exposure, and vulnerability. We agree also with point s (a, b). The risk definition proposed in our document has been, at the best of our knowledge, firstly introduced by Crichton in 1999. The definition and the references have been accordingly updated. "Object of the risk analyses" have been replaced, as suggested, by "elements at risk". See lines 70 to 75.

**Comment 4**

When summarizing the "current" state of the literature, a reference published in 2004 should be complemented by more recent publications (in lines 77-78).

**Reply**

We appreciate the suggestion to include more recent studies, and we totally agree. Sometimes we had to go back to less recent references to find support for some specific topics. We updated the sentence with a more recent reference. See line 91.

**Comment 5**

The authors are encouraged to revise the enumeration which presents "contingent valuation" and "willingness to pay" approaches as a list suggesting that they are different approaches – it could

be clarified that WTP approaches fall under CV methods (as mentioned in the reference provided by the authors) (lines 82-83).

**Reply**

The sentence has been updated, emphasizing that the "willingness to pay - accept" falls under the Contingent valuation methodology as correctly noted. See lines 95, 96.

**Comment 6**

The authors could use the introduction to provide more background on previous research on the topic of environmental damage to environmental assets. The single reference that was cited (Tait 2019) should be added to the bibliography. It is a non-academic publication. If possible, adding some academic references to position the proposed paper in the broader academic landscape could help to highlight the contribution of the proposed paper to the existing literature. E.g. there is research on damage to ecosystems from flooding. The proposed study is intending to expand

**Reply**

Thanks for commenting on this. We decided to present the topic of environmental damage without going in deep of it, since our study is focused only on the exposure. But we understand your comment and we agree that we can expand the background provided on previous research related to environmental damage to environmental assets, adding academic references where possible. Tait has been added in the bibliography.

The introduction has been revised following the hint provided by your comment. Several references have been added in support of the potential vulnerability of environmental assets to floods. We think that now the framework is clearer and better presented to the reader. Moreover, the scientific background is now sounder. See lines 34 to 56.

**Comment 7**

The rational for selecting a limited geographic scope (Europe and Italy) should be provided. The reading flow would benefit from an explanation why these regions are especially relevant (lines 51, 55, and 139 mention Europe & Italy, without explanation of why these regions were selected).

**Reply**

The framework in which the work has been developed was clarified, with the modifications at lines 62 to 64. Clarifications have been added also at line 133. We believe that both the reading flow and the clarity of the paragraph benefit from these modifications.

**Comment 8**

The authors could clarify what they refer to (asset types and geographic scope) when saying "all the assets" in line 142.

**Reply**

The sentence has been modified ensuring that the scope of asset types and geographic coverage is now clearly communicated. See lines 174, 175.

**Comments to "Materials and methods" section**

**Comment 1**

In section "2.2.1 Level 1," the explanation (lines 221-229) would benefit from an example, e.g. providing a map with differently weighted areas and the calculation of the score. In the current version of the text, it is not very clear how the values are assigned. What value is assigned to e.g. an UNESCO heritage site or to a Ramsar site?

**Reply**

We agree that the explanation in section 2.2.1 would benefit from a practical example, and it has been integrated with at least one example for each weight, and according to us, the weight assignment procedure is now clearly exemplified, also without adding a map here, since we would like to keep maps only for the results section. See lines 285 to 290.

**Comment 2**

Section 2.2.2 Level 2 mentions "woods" as an environmental asset. In ecosystem services assessments, usually "wood" is be considered a forest product, hence it is counted under provisioning services of the forest. However, the ecosystem is the forest. The authors could explain what categories are used to describe "environmental assets" before presenting the weighting approach. The authors could further place the method proposed for the weighting of ecosystem services in the context of existing research that uses weighting approaches for ecosystem services, e.g. multi-criteria decision analysis for nature conservation.

**Reply**

The issue here is due to the use of "woods" as synonym of "forests", and this can generate misunderstandings. Thanks for noticing it, the word "woods" has been substituted with "forests". See line 301. Regarding the classification and the categories, they are not presented here since we prefer to strongly divide methodology from results. We hope that overall modifications contribute enough to the clarity of the manuscript. A search regarding multi-criteria decision analysis has been conducted, and a paragraph illustrating the framework supporting the decision of using weights has been added to the text, including literature support. See lines 311 to 316. Thanks for suggesting the nature conservation as a field in which finding literature support for our weighting method.

**Comment 3**

The formula presented in line 275 would benefit from further visualization of the structure used to order the ecosystem service subcategories. The authors could clarify based on which criteria the 16 ES were selected.

**Reply**

We clarified the criteria used to select the 16 ecosystem services (ES) categories, providing a more detailed explanation of the structure used. See lines 339, 340. To improve the reading of the structure of the subcategories, the formula has been substituted by a graphical explanation of the adopted structure. See figure 2 at line 343.

**Comment 4**

In the section "2.3 Survey" the methodology description shifts from a general methodology overview to a specific survey conducted in Italy. The authors could consider aligning the framing of the different methodology sections. One way to do that could be to present a general methodology first and then apply it to the case study area or, alternatively, introduce the case study area (and choice selection) first and then apply the proposed methodology directly to that area. It is recommended to present the full formula to get to the result in line 315.

**Reply**

The survey has been developed without focusing on the Italian audience, but has been administrated mainly, but not exclusively, to an Italian public. But the survey is not limited to the case study, which is a separate section of the manuscript. We rearranged the description removing the reference to the case study, see lines 374 to 379, to clarify this aspect. The point should now be solved. Clarifications about the weighting formula have been added, in line 388, 390, 398, and then the formula at line 400, now fully explicating the process of the weight assignment.

**Comments 5, 6**

While the objectives of the paper are clearly stated, the methodology would benefit from a clear statement of the research question.
The methods section could state more clearly which maps are used to identify the environmental assets: 1. in general, for future studies using the proposed methodology and 2. for the Italy case study.

**Reply**

We pointed out in the manuscript where we present the elaborations made for fulfilling the research objectives presented in the introduction. See lines 146, 183, 194. An additional clarifying sentence has been included in the paragraph 2.2. Lines 182 to 184. For collecting all the assets, a variety of maps and datasets have been accessed and used. Those come from the official datasets released within the laws and other datasets from public authorities. The list is provided as supplementary material. We refer to it at line 428. Following all the suggestions, we clarified in different sections of the manuscript that for transferring the methodology to different areas, internationally recognised assets will not change, but for local assets specific studies may be necessary. The detailed explanation is provided from line 161 to 167. The manuscript as structured after the suggested revisions should now be clear with this point.

**Comments to "Results and discussion" section**

**Comment 1**

I haven't fully understood yet what the benefit of the proposed typology is compared to existing typologies of ecosystems, e.g. the red list of ecosystems. The paper would become much stronger if the process of creating the list of assets would be explained in more depth. "MTB tracks" seem to be listed as a single category, while hiking tracks are not. If that was a deliberate decision, it might be good to explain the decision. The site "Gas emitting" seems to be missing a noun.

**Reply**

The proposed typology differs because it relies on the assets that are individuated by the legislative framework of the place in which the assessment is conducted. This helps because a clear selection of the assets could be tricky or less applicable if they are not recognised by the local authorities. For this reason, we also highlighted that for comparing two regions, the assessments must be done on the same assets, because they may change (depending on the legislative framework, they can be unified across a nation, or some can vary among different regional administrations, as it happens in Italy). MTB tracks are a separated category in the protection laws of our case study, while no distinction is done for hiking trails. Anyway, the taxonomy has been revised also to fulfil the requests expressed in the other comments and more details about the overall modifications are provided in the reply to the next comment. Gas emitting was corrected adding "sources".

**Comment 2**

While before, the reference case is Europe or Italy, the taxonomy aims to represent protected environmental assets in Europe and internationally (line 371). It should be discussed to which degree the selected case can represent the whole world. It could be clarified under "3.2 survey results" (or before) what the geographic coverage of the survey was.

**Reply**

Due to the clarifications asked for the taxonomy, we decided to update it improving its transferability and keeping it well separated by the case study. See the new taxonomy presented in figure 4, line 467. The full explanation has been included in the manuscript, from line 451 to 465. This modified classification and explanation, made following your concerns, should have clarified all the potential misinterpretations.

**Comment 3**

Types of areas under consideration: In the introduction, it was not highlighted that the focus is on "protected" environmental assets. Section 2 on methods mentions that "they are usually protected by national or regional laws, which can be used as identification instruments" (line 138f.). The authors identify all protected assets in Europe (and Italy?) and based on this derive a typology of environmental assets. In the results, the focus is only on protected areas. There are non-protected ecosystems that provide considerable ecosystem services. The authors could explain why they limit their analysis to protected areas.

**Reply**

All the integrations apported so far, should have resolved the unclear point of which asset we are considering, particularly, see lines 159 to 167.

**Comment 4**

After reading the paper, it seems like the case study focuses on riverine flooding. This specification could be made in the introduction.

Gabriele Bertoli, Chiara Arrighi, Enrica Caporali

**Reply**

Yes, we focused on riverine floods in the case study, we now stated it clearly, adding the word "river" in lines 18 and 37.

**Comment 5**

The wording "natural events" (in line 569) could be misleading in times of climate change, where flood events are becoming more severe and more frequent and hence are not only "natural" but also also driven by anthropogenic climate change. It might be more suitable to say flooding instead of "natural events."

**Reply**

That definition is referred to a non-correct, unfortunately common, practice. We don't focus on the climate change effects on floodings in this study, but we agree that most of the damages potentially induced by flooding to an environmental asset are human derived, e.g. pollution. We substituted the word "floodings" with "natural events", as suggested. See line 670.

**Comment 6**

In conclusion, the authors are encouraged to consider the literature on flooding-induced damage to ecosystems and on methods for weighting ecosystem services when presenting the existing literature in order to place the proposed study within an existing body of literature. The paper could benefit from explicit statements regarding the decisions made in the research process (choice of case study area, choice of maps to generate the environmental asset taxonomy). Finally, the authors are encouraged to discuss the generalizability of the taxonomy, considering that the case study is limited to the European context.

**Reply**

Thanks for the suggestions for improving our manuscript. We revised the conclusions chapter, adding several sentences and references backing all the changes made so far, and better supporting the choices done during the study. See the integrations regarding the taxonomy in line 657, 658. References to better position our work in the literature have been added from line 673 to line 682. The survey section has been rewritten to take in considerations the improvement suggested by the comments and to clarify the strengths. See lines from 694 to 708.

**Comment 7**

I see the strengths of the paper mainly in achieving objectives 2 and 3, which are filling a gap in the existing disaster risk assessment research. The authors could consider framing these as the key contributions of the paper.

**Reply**

We thank you again for the dedication you applied in revising our manuscript, strongly contributing to making it better presented and structured. Although, we are convinced that also the objective 1, "identify what should be considered as environmental asset in a flood exposure analysis, i.e., define a taxonomy for exposure", is fully achieved in the work, and with all the modifications applied from the above comments, this issue is believed to be resolved. We believe that the work benefit from your suggestions and is now clearer communicated. The sentences

regarding the objectives have been updated in the discussion and conclusions section. See lines 650, 661.

**Minor comments**

The authors could try to simplify the language when describing the objectives, e.g. the objective "identify what should be considered as environmental asset in a flood exposure analysis, i.e., define a taxonomy for exposure" could be summarized by saying "develop a taxonomy for environmental assets exposed to flooding." The objective to "develop a new method for valuing the environmental assets able to differentiate among asset typologies, and which is not directly based on the economic value of the asset" could be shortened by saying "assessing monetary and non-monetary values of environmental assets."
In some instances, the grammar could be improved and prepositions could be updated, e.g. "contaminants originated by human activities" should be replaced by "contaminants originating from human activities." This could be achieved by using an AI tool for a quick grammar check.

**Reply**

Research objectives have been updated, see lines 129 to 132.
Where possible (without changing the meaning of the sentences), we simplified the language and improved the grammar. Thanks.

Gabriele Bertoli, Chiara Arrighi, Enrica Caporali